# The Impact of Virtualisation Techniques on Power System Control Networks

**Friederich Kupzog [1],\*, Armin Veichtlbauer [2] , Alexander Heinisch [3],**
**Ferdinand von Tüllenburg [4], Oliver Langthaler [5], Ulrich Pache [5], Oliver Jung [6], Reinhard Frank [7]**
**and Peter Dorfinger [4]**

[1] Center for Energy, AIT Austrian Institute of Technology GmbH, 1210 Vienna, Austria
[2] Research Campus Hagenberg, University of Applied Sciences Upper Austria, 4600 Hagenberg, Austria;
    armin.veichtlbauer@fh-hagenberg.at
[3] Siemens AG, Corporate Technology, 1210 Vienna, Austria; alexander.heinisch@siemens.com
[4] Salzburg Research Forschungsgesellschaft mbH, 5020 Salzburg, Austria;
    ferdinand.tuellenburg@salzburgresearch.at (F.v.T.); peter.dorfinger@salzburgresearch.at (P.D.)
[5] Center for Secure Energy Informatics, Salzburg University of Applied Sciences, 5412 Puch bei Hallein,
    Austria; oliver.langthaler@fh-salzburg.ac.at (O.L.); ulrich.pache@fh-salzburg.ac.at (U.P.)
[6] AIT Austrian Institute of Technology GmbH, Center for Digital Safety and Security, 1210 Vienna, Austria;
    oliver.jung@ait.ac.at
[7] Siemens AG, Corporate Technology, 80333 Munich, Germany; reinhard.frank@siemens.com
\*   Correspondence: friederich.kupzog@ait.ac.at

**Abstract:** Virtualisation is a concept successfully applied to IT systems. In this work, we analyse how virtualisation approaches, such as edge computing, brokerage and software-defined networking, can be applied in the area of electricity grid management and control systems. Power system information and communications technology is currently subject to significant changes. Networked power grid components including renewable energy units, electric vehicles and heat pumps need to be integrated into grid management systems. We studied how virtualisation techniques can support system operators in increasing an energy and communication system's dependability and situational awareness, and how manual (mostly field-level) configuration and engineering efforts can be reduced. Starting from a working hypothesis, three concrete use-cases were implemented and the performance enhancements were benchmarked to allow for well-informed answers to the questions above. We took a close look at application-protocol-independent redundancy, grid-based routing and online system integrity control. In these study cases, we found significant improvements could be achieved with virtualisation in terms of reduced engineering effort, better system management and simplification in high-level system architecture, since implementation details are hidden by the virtualisation approach.

**Keywords:** network virtualisation; software-defined networking; redundant device deployment; feeder configuration; anomaly detection; reliability; smart grid; grid automation; programmable connectivity

---

## 1. Introduction

Information and communication technologies (ICT) play a key role in the integration of renewable energies into existing power grid infrastructures. Applications such as distribution grid monitoring, control of field devices and distributed energy management are becoming increasingly important alongside conventional applications, such as supervisory control and data acquisition (SCADA), metering and billing. With worldwide efforts to reduce carbon dioxide emissions [1],

new requirements for existing energy systems are evolving, such as the need for sector coupling and efficient implementation of energy communities [2]. This includes the integration of a large number of new system components either into the core infrastructure control systems, or into networks of grid-connected entities such as energy communities or virtual power plants.

Under these circumstances it seems grossly insufficient to merely scale up the existing ICT systems of today's distribution grid operation and enhance them with a state-of-the-art security concept [3]. Currently applied procedures for tasks such as outage management, configuration of new network components and testing of new IT network segments are typically performed manually and become highly inefficient when scaled.

Virtualisation concepts from the ICT field, such as edge and cloud computing, dynamic virtual local area networks and software defined networking (SDN), open up a potential opportunity to address this scaling challenge in today's power system ICT [4–6].

In this work, we analyse how and how well virtualisation concepts from the ICT domain can improve relevant power distribution use-cases. We focus on the application of communication network virtualisation in cases such as configurations of new protocol stacks, cross-layer optimisations between energy and communication networks, the integration of non-IP traffic and legacy components and online integrity checks of energy and ICT systems. In the virtualised solutions, components of a distributed control system can be configured and operated at a virtual central location.

In order to systematically assess the potential benefits of virtualisation approaches in the context of power system ICT, we studied the following three use-cases and implemented new solutions for them:

*Application protocol independent redundancy*: Virtualisation supports the openness of existing infrastructures for new protocol stacks and the integration of non-IP traffic, including legacy protocols. Existing infrastructure can be routers, switches or customer premises equipment. Virtualisation-enabled networks spanning these components can adjust to altered protocols by simple, centrally managed software changes. In today's power system automation systems, dedicated hardware is required to support special protocol suites, such as IEC 60870-5-104 or IEC 61850. With network virtualisation approaches, ICT infrastructure can easily be adapted to any protocol stack. A use-case where the protocol-specific configuration is relevant is the setup of redundant components, e.g., controllers, in a power system substation. Usually, the switch-over from one controller to the other is implemented using dedicated parts of the IEC 60870-5-104 protocol and requires significant engineering effort regarding configuration. By proposing an application-protocol-independent solution using OpenFlow, we aim to significantly reduce the required effort.

*Cross-layer optimisation and grid-based routing*: With better control over the communication network configuration by means of virtualisation, it becomes easier to closer integrate application functionality with networking aspects. We studied the routing of measurement information from localised sensors to location-specific controllers. In a power distribution network, the association between sensors and controllers changes with topological changes of the grid, e.g., initiated by feeder configuration changes due to switch operations. In this case, application logic, network logic or a combination of both has to re-arrange the source-destination binding for measurement data and potentially also control commands. We compare the benefits of different virtualisation approaches for this application (brokerage, OpenFlow, Programming Protocol-independent Packet Processors (P4) (https://p4.org/) and propose solutions that route these measurement data to the appropriate destinations (based on the actual grid state) without any reconfiguration of the measurement units.

*Online system integrity control*: Proactive detection of failures and cyber attacks in ICT networks and potential conclusions regarding anomalies in the power system operation are in focus of this use-case. The detection of attacks in combination with adequate countermeasures results in higher system availability. We studied the implementation and associated restrictions for domain-specific anomaly detection in power system control networks when using the northbound interfaces made available by SDN controllers.

## 2. Research Questions and Choice of the Methodology

To systematically evaluate the impacts of virtualisation techniques on power system control networks, this work analyses the following three research questions:

1.  Which approaches allow for minimising configuration efforts and manual engineering in patch management when integrating large numbers of new, intelligent power grid components?
2.  How can system reliability be improved and graceful degradation be realised using the re-location of distributed control information in cases of ICT malfunctions or even connection loss?
3.  How does SDN support situational awareness in power system ICT networks, including proactive detection of overload conditions, malfunctions or malicious attacks?

A research design appropriate to answer the research questions can be chosen in different ways. One option is the use of a model-based system engineering (MBSE) approach [7]. In MBSE, a formal model of the system of interest is developed and can be used for code generation, test case generation or automatic verification and more. This approach would allow a structural evaluation of the advantages of the virtualisation-based problem solution by programmatic analysis of the model. In the model-driven architecture (MDA), the prime artefact of the work is actually the model itself, which is typically formulated using Unified Modelling Language [8]. Model-driven engineering (MDE), which makes use of domain-specific languages [9], has already been successfully applied to the power system domain already [10].

However, the research questions address a range of mostly qualitative questions. This is due the nature of the initially open question about the potential advantages of virtualisation in power system control networks. Here, a formal system model based on a domain-specific description language comes with challenges. The scope and application environments of the use-case implementations were unknown in the beginning of the research, and with this the structure and potential complexity of the model. Therefore, this work is based on an empirical approach. We conducted applied research and implemented three study cases that allow us to generalise the findings to a certain extend. With that, we established the validity of our claims on the "accumulated weight of empirical evidence" [11]. This approach is also known as "validation by example" in the taxonomy of Shawn et al. [12]. Nevertheless, we describe our approaches and results in the context of a high-level architecture model, as described in Section 3, to better show the similarities in the virtualisation of the three study cases.

The applied methodology consists of the following steps: First, practically relevant use-cases for virtualisation in power system control networks were identified by a team of power system operators, solution providers and researchers. Second, a high-level model for power system control systems with implemented virtualisation techniques was set up (this is described in Section 3 and shown in Figure 1). This model is a high-level architectural view and allows one to generalise the virtualisation of networking functions in the given application context. We claim that virtualisation reduces engineering effort in the practical deployment and maintenance of the identified use-cases. Third, to support the claim, for each of the three use-cases, a theoretical solution was developed and evaluated: how are planning and operational processes changed by the solution and what are the implications of these changes? The fourth step was the experimental implementation of promising approaches in all three cases. For the evaluation, experimental test-beds in simulations, the laboratory and a real field environment were set up, where the initial hypothesis of feasibility and the impacts of the virtualisation approaches could be validated.

The remaining paper is structured as follows:

Section 3 presents the model for power system control systems with the implemented virtualisation techniques and the hypothesis of this work.

Section 4 analyses the state-of-the-art in power system control network management and virtualisation techniques, and gives insights into the situation in the field today.

Section 5 discusses the applicability of network virtualisation techniques to power system control networks, in general, and the three use-cases as presented in the introduction, in particular.

Section 6 presents the use-case implementations based on the applicability analysis.

Section 7 discusses insights from the practical implementations and answers the above research questions and how the initial hypothesis is supported by them.

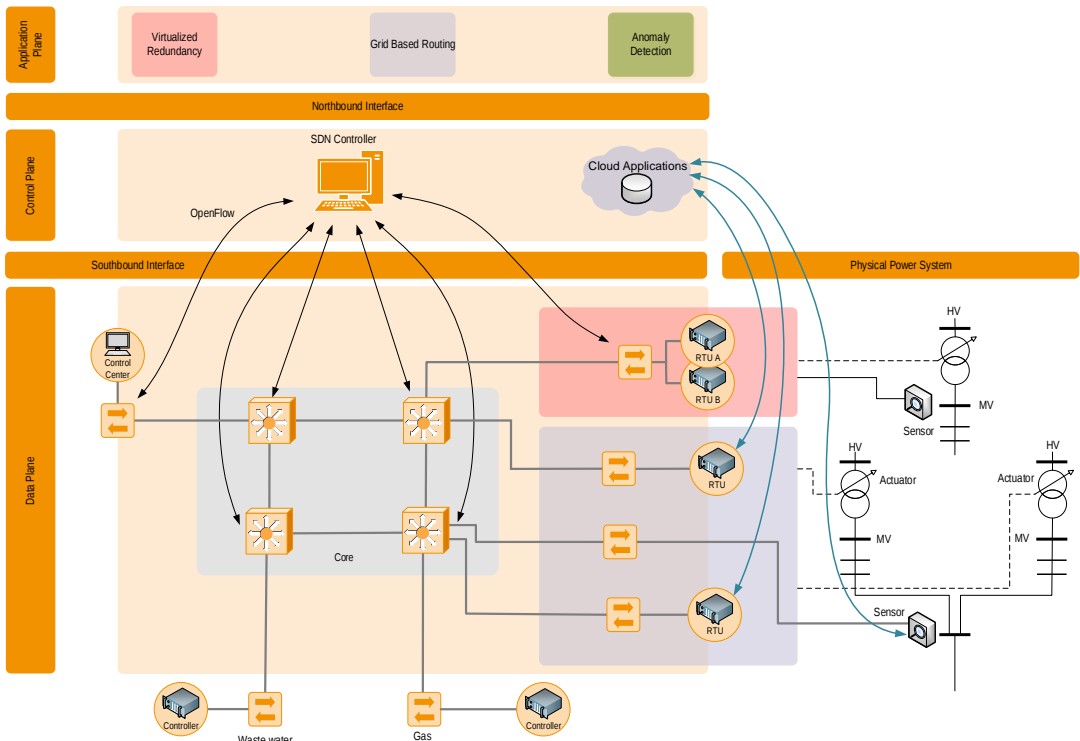

**Figure 1.** System model for communication virtualisation in power system control networks.

## 3. System Model and Hypothesis

In this work, we analyse how engineering effort and system complexity can be reduced by applying (network) virtualisation. A model of the studied system is shown in Figure 1. The communication network components can be found on the left, and the ICT-connected power grid components (sensors and actuators), such as on-load tap changing transformers and voltage measurement devices, are located on the right. Moreover, a typical multi-service utility also used the communication network for the gas distribution system and water supply system, as shown in the bottom of the model.

We assume that at least parts of the communication system are SDN-enabled. For this, at least the first and last router of an SDN-enabled communication route must be SDN routers. For better clarity, we summarised the different network parts used in practice (core network, substation network, etc.) into one single data plane. The SDN(s) come with their dedicated SDN controller shown in the control plane of the model between its southbound interface (towards networking components) and its northbound interface (towards application plane).

As alternative virtualisation approach to SDN, we also consider P4, cloud and edge computing (see Section 5). SDN and cloud approaches are shown as examples in Figure 1. In the model, these approaches are on the same conceptual level as the SDN controller, allowing access to an underlying infrastructure and virtualising its distributed nature. These two techniques are able to support a broad range of use-cases, of which this work studies three examples in detail (application-protocol-independent redundancy, grid-based routing and online system integrity control).

The hypothesis of this work is that they can significantly reduce the complexity of use-case implementations on the application plane and thus also the engineering effort required. The different uses cases can be implemented as separate SDN or cloud system applications on top of the northbound interface. In Figure 1, the representations of these applications in the data plane are coloured boxes. We argue that the engineering effort implementing use-cases on the application plane can be reduced using virtualisation techniques because the control-plane layer encapsulates the complexity of the lower-level systems (data plane and physical power grid).

While the application-protocol-independent redundancy (red) and grid-based routing (violet) use-cases have dedicated representations in the data plane, the anomaly (light green) use-case is in principle interacting with all SDN switches in the network. The core network switches are SDN switches that are controlled by a central SDN controller.

## 4. State of the Art

Communication networks play an important role in the context of SCADA systems. Sensor values have to be transported from field devices to the control systems and control commands need to be transmitted in the opposite direction. Traditionally, power system operators maintain dedicated communication networks for their control and monitoring systems. These communication networks follow three major design goals: scalability in terms of the number of connected assets, operational safety in terms of robustness against failures of the communication network and interoperability with respect to the diversity of connected devices.

Following that, the control and monitoring networks are typically based on standard technology. On the physical layer, the wired networks are usually based on copper and (increasingly common) fibre-optics, and cable trajectories typically follow the power line routes. Additionally, power line communication is frequently used. Slightly less common are wireless connections using directional radio. Here, the radio stations are usually located at particularly exposed locations owned by the network operators. In order to fulfil the requirement on operational safety, the networks typically form a ring-of-rings topology (see Figure 2). In this way, each communication end point is reachable by least two paths. In order to ensure interoperability and scalability from a communication protocol view, the networks are typically operated using proven layer-2 network technology [13].

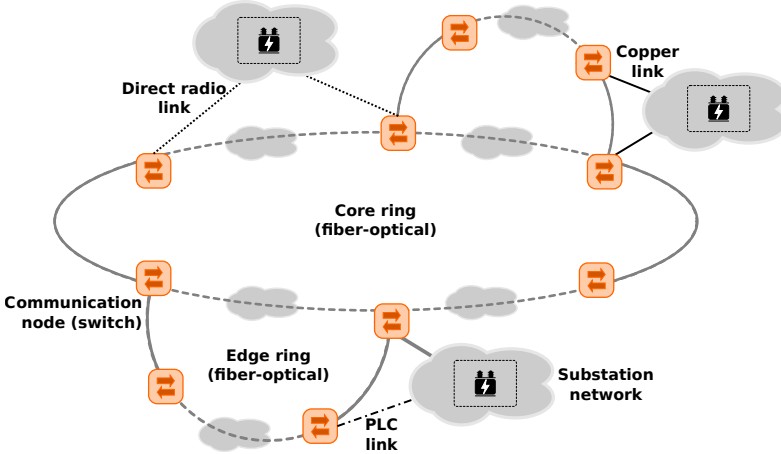

**Figure 2.** A schematic illustration of a typical ring-of-ring topology, consisting of a core ring and multiple edge rings connecting the field stations of remote grid areas. The rings are often based on fibre-optic technology, while the spoke links to field stations may use different layer-1 technologies.

### 4.1. Conventional Communication Technologies Used in Power System Control Networks

Communication networks span across large geographical areas and interconnect large numbers of nodes, which are part of a diverse set of applications. From the view of network management, solutions are required which support the operation of the network with regard to the initial design goals.

Virtual (extensible) local area networks (VLAN/VxLAN) provide a common approach for dividing the overall layer-2 domain into smaller virtually separated segments. These segments contain a subset of the available links of a network and interconnect a subset of the connected communication nodes. In that way, VLANs are commonly used to separate the data flows of different applications, and thus, minimise mutual influences between different applications. VLAN uses the Ethernet extension IEEE 802.1Q which allows up to 4096 network segments. However, together with the growth of data centre and cloud structures, the VLAN segment limitation has turned out to be problematic. Hence, the VxLAN overlay technology has been developed to overcome the scalability issue of VLANs [14]. Additionally, VxLAN allows one to interconnect multiple LANs across wide area networks (WANs), such as the Internet, in a more simplified way using encapsulation. VxLAN endpoints open an overlay network across the WAN network (e.g., the Internet), and LAN packets are tunnelled between the LANs transparently.

While VLAN and VxLAN have been rolled out widely in monitoring and control networks of energy systems, the decentralisation of the energy systems led to a progressing digitalisation and a change within the control and monitoring procedures (grid codes). As a result, new requirements arose that cannot be easily fulfilled with VLAN or VxLAN. This holds true in particular for increased dependability of the control and monitoring data flows between SCADA systems and field devices. These data flows have particular demands on the communication quality. For instance, some monitoring functions such as wide area monitoring systems (WAMS) require tight time-synchronisation of sensors (e.g., for phasor measurement units—PMUs) and some protection functions (e.g., line tripping) have strict demands on the communication latency and packet loss. Such requirements cannot be fulfilled by VLAN and VxLAN, and thus, multiprotocol label switching (MPLS) is increasingly entering SCADA networks. The MPLS protocol provides protocol independence, network resource management and failure mitigation. The basic principle behind this approach is packet labelling. Data packets belonging to a certain application or data flow are provided with an additional header containing a label identifying the application or data flow the packet belongs to. The forwarding nodes (MPLS router) within the network enforce particular treatments for these packets based on configurable rules. For instance, these rules define over which link the packets of a certain label are being forwarded, or which portion of a link's bandwidth is reserved for packets of a certain label.

MPLS allows one to define particular communication quality parameters for different data flows by strictly governing the routes and bandwidth a certain flow is assigned within the network. For this, MPLS has been extended with a traffic engineering (MPLS-TE) function. Additionally, extensions of the MPLS protocol provide path redundancy through a fast re-route mechanism (MPLS-FRR). This mechanism allows the protocol to switch between redundant paths as soon as one of the paths fails. Compared to the traditional rapid spanning tree protocol, switch-over times can be significantly reduced in order to meet latency requirements.

### 4.2. Software-Defined Networking

The SDN concept is based on a networking design philosophy that advocates the separation of the network data plane from the network control plane, and thus facilitates programmatic network management and enables efficient network configuration (see Figure 3). Changing the network configuration in non-SDN networks requires significant effort and time. With the availability of the SDN programming interface, network elements can be updated from the central controller (i.e., at the control plane) and complex network management tasks can be carried out more easily. SDN software interfaces provide more flexibility for network management and control, allowing for fine-grained traffic engineering [15,16]. Changes to the network can be made collectively without requiring a device-centric configuration. SDN moreover provides network virtualisation capabilities and enhanced security through the visibility of all network components to the SDN controller.

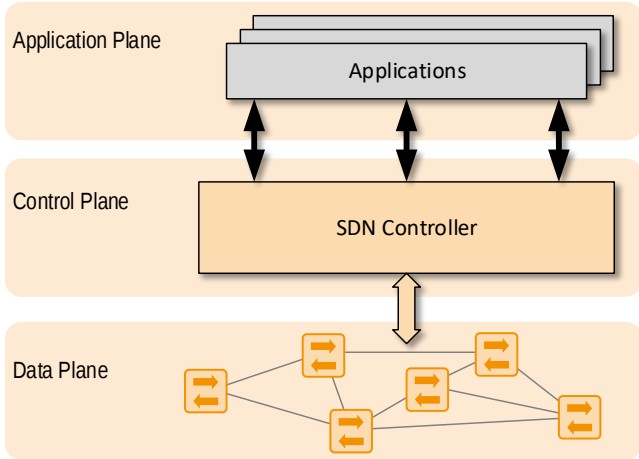

**Figure 3.** SDN architecture.

Plenty of research has been conducted on the use of SDN in smart grids. The use of SDN has been found to be beneficial for smart grid communication. Rehmani et al. [17] lists some main motivations for the use of SDN in smart grids.

- Separation of traffic of different traffic types.
- Quality of service.
- Virtual network slicing.
- Enhancing resilience of smart grid communication.
- Fast failure detection and recovery.
- Timely load shifting to prevent voltage collapse.
- Interoperability and easier network management.
- Reduction of management complexity of the integration of electric vehicles.

Concerning resilient smart grid communication, Aydeger et al. [18] proposed an inter-substation communication network based on SDN in order to enable inter-substation communication. The approach employs a hierarchy of SDN controllers: the global SDN controller manages the traffic between the substations and is deployed in a central location of the grid operator. Local SDN controllers are deployed in the substations to manage local traffic. Resilience is increased through the use of redundant links whenever it is required.

The self-healing capabilities of smart grids can be improved by using SDNs to reroute traffic of compromised PMUs as shown in [19]. PMUs are used to monitor the voltage-level and phasor angle of transmission lines. Phasor data concentrators (PDCs) receive the data from various PMUs and send the collected measurements to the control centre. If a PDC is compromised by a cyber attack, the monitoring capability of the smart grid is severely impacted if not lost. This problem is mitigated by rerouting traffic from PMUs to uncompromised PDCs using SDN.

Dorsch et al. [20] proposed a fault tolerance mechanism for SDN-based smart grids by minimising the time for link failure detection and link failure recovery and optimising paths after link recovery. To detect link failure, they used bidirectional forwarding detection and heartbeat mechanisms of the SDN controller. For link recovery, they used fast failover groups, which are a part of OpenFlow. The SDN controller was used for post recovery path optimisation. Links with lower load were selected over links with higher load in order to achieve optimal performance of the new paths.

Fast failover groups were also used by Pfeiffenberger et al. [6] to increase reliability in substation multicast communication by reducing the number of lost packets between link failure and recovery.

The SDN controller has a complete overview of the network. This can be used to effectively forward traffic through the SDN along optimal paths. Montazerolghaem et al. [21] proposed the OpenAMI routing scheme to find the shortest routes and load balance the traffic through an advanced metering infrastructure (AMI) network. Through the use of OpenAMI, low end-to-end delay and higher throughput could be achieved.

The complete overview of the network, as provided by an SDN controller, is also used to enhance the security of smart grids. In [22], a network-based intrusion detection system for SDN-based SCADA systems is introduced. This system relies on SDN to monitor the communication between power grid components, to capture network information and to periodically gather network statistics. The collected data are used in a one-class classification algorithm to detect malicious traffic.

Zhang et al. [23] claim that the deployment of SDN can bring tremendous benefits for smart grid communication networks. They highlight in particular, SDN capabilities such as simplification of network configuration and management, cross-domain content-based networking and virtualisation and isolation that can offer considerable synergies between the smart grid and SDN.

Cahn et al. [24] proposed using SDN to tackle many problems that arise in substation communication networks. They introduced a software-defined energy communication network (SDECN) architecture with auto-configuration capabilities that can eliminate many known issues in substation management. They argued that, due to the vast number of different intelligent electronic devices (IED) that monitor and control a dedicated part of the substation and mainly use layer-2 communication, network configuration is complex. The SDECN can significantly reduce required effort and support the configuration of layer-2 multicast groups and the deployment of new IEDs in the network.

SDN features such as improved network visibility and easy re-configuration can be used to implement sophisticated network security applications and traffic isolation that help in improving network security. However, the central SDN controller increases the attack surface. The topic of SDN security was summarised by [25,26].

### 4.3. Programming Protocol-Independent Packet Processors (P4)

Although SDN introduces programmability on the control plane, OpenFlow switches still have a fixed-function data plane. As OpenFlow only supports certain protocols and header fields, controller operation is limited to flow table management and processing of packets received from the OpenFlow switches. This follows a bottom-up approach where SDN developers have to build the control plane on the basis of the fixed functionality of an SDN-capable data plane.

P4, however, provides programmable data planes that allow data plane behaviour to be defined on an abstract layer. Thereby, both the control and data plane become programmable.

Programmable data planes provide basic functions, e.g., match-action-tables and mechanisms for header manipulation, that can be configured and chained to form a specific forwarding pipeline. The specification of packet processing becomes independent of the switching hardware; i.e., it is portable across hardware targets.

This flexibility introduces many advantages. Development of networking hardware can be decoupled from network function programming. Instead of being limited to long-cycle hardware development processes, new functions can be developed and deployed easily by developers. In addition, bugs in the forwarding pipeline can be fixed without the need to wait for updates. Therefore, programmable data planes simplify agile development processes with easy prototyping, rapid design cycles and simple deployment.

The P4 technology is expanding in the data centre area and in service provider networks. Nevertheless, it is also relevant for smart grid networks in certain places to solve dedicated local problems. Programmable data planes, as provided by P4, can improve the delay values with hardware-encapsulation methods for a site-to-site tunnel architecture. Moving complex and

computation-intensive processes such as Internet Protocol Security tunnelling to hardware via a programmable forwarding plane will result in optimised site-to-site connectivity.

As shown in Figure 4, the P4 architecture allows one to analyse, modify and introduce each type of packet header, standardised or proprietary. The headers or information fields can be introduced everywhere in the data frame. This offers the flexibility to introduce additional information into data packets not via software processes but instead in hardware and thus with wire-speed on specific hardware targets.

**Figure 4.** P4 architecture.

## 5. Virtualisation Techniques and Their Applicability in Power System Control Networks

The aim of this section is to give an overview of existing virtualisation concepts in the ICT domain and to assess their applicability to the mentioned use-cases (application-protocol-independent redundancy, grid-based routing, and online system integrity control). A pre-selection has already been made throughout the state-of-the-art analysis at an earlier stage of the research work, resulting in the following virtualisation concepts to be considered for the implmementation of the named use-cases: VLAN/VxLAN, MPLS, cloud and edge computing, SDN, and P4. Based on the applicability of these technologies to the use-cases, a common meta-architecture has been developed.

Virtualisation may take place at several places in a comprehensive ICT infrastructure. All of them have to be considered in order to find the best solutions for the named use-cases and to derive an appropriate meta-architecture. In the following, a short overview of possible approaches and how well they fit the expectations regarding the use-cases' requirements is given.

### 5.1. Device Virtualisation

First of all, the simplest approach is to virtualise physical systems. These can be end systems taking part in the smart grid, such as IEDs, or intermediate systems as part of the communication infrastructure. Some of these devices may well be realised as virtual machines (e.g., back end servers running on a hypervisor), whereas others require physical interaction with the real world environment (e.g., sensors and actuators in the field), in which case they cannot easily be re-located.

When using public communication infrastructures such as the Internet as an underlay infrastructure, the participating active network components (routers, switches) are also hard to get rid of, as the packets have to be forwarded by some kind of device. However, virtualisation in networked environments is not uncommon [27]. In its simplest form, virtual links are defined by technologies such as link aggregation, where a number of physical links are bundled to form a logical link with greater capacity.

Another very common technology is the definition of virtual local networks (VLANs), which are logically separated IP networks within a switched infrastructure. Furthermore, some networks are

totally virtual, i.e., not existing in real infrastructures (e.g., VLANs of a virtual network environment hosted within a hypervisor). Thus, the participating switches are realised as virtual switches ("vswitches"). In reality, a co-existence of real and virtual infrastructure is very common [27].

The idea of virtual machines (VMs) allows for the flexible distribution of these VMs onto the existing hardware. Changes of the real hardware do not necessarily affect the VMs, as long as the definition of the requirements for the respective VMs can be satisfied by the real infrastructure. Conversely, additional VMs may be run on the same hardware whenever required, as long as the performance of the underlying infrastructure is sufficient.

Changes in the control process of electrical systems (be it due to regulatory changes, due to changing requirements by the electrical part of the infrastructure or for any other well-founded reason) can thus be mapped onto the same physical infrastructure. This makes device virtualisation especially useful for redundancy-grounded use-cases such as application-protocol-independent redundancy or grid-based routing.

In networking, an abstraction of hardware resources into logical parts, which are requested via service level agreements, has been introduced and has received increasing interest in the last few years. This technology is known as "network slicing" [28], wherein the slices are portions of the physically available bandwidth, which are associated with a requesting user or application. At the moment, the focus is in the area of 5G cell-based radio communication. This makes it interesting for all applications in the low-voltage grid, since in many cases, no fixed-line communication is available.

*5.2. Functional Virtualisation*

However, virtualisation can also be considered at a purely functional level. From this viewpoint, it is not important where, but under which quality conditions functions are performed (in which time, to which degree of correctness, etc.). This is often referred to as cloud computing [29], as the location is not a decisive factor (which does not really hold true for edge computing [29], as the location "at the edge" of the public network has an important influence on features such as trustability and real-time capability).

This functional virtualisation may be used for each kind of service which is needed within the ICT solution. It may apply to applications such as billing, where the functionality is provided by cloud software ("software as a service"). In this case, questions of liability have to be sorted out in advance—again, usually by defining legally binding service level agreements.

The same also applies to network functions ("network function virtualisation"). For instance, routing (application layer routing) or security (intrusion detection) functions may be used in a virtualised way. Often, these virtualised functions are bound to virtualised devices, which may exist in cloud or edge environments ("platform as a service"), or are part of a complete virtual infrastructure ("infrastructure as a service").

This feature is especially beneficial for the online system integrity control use-case, as anomalies in the network infrastructure can be easily detected by such virtualised functions. This detection is not only based on intrusion signatures, but it may also cover unknown outliers regardless of their origin. Such anomalies may be produced intentionally by some attackers, but also unintentionally by improper resource usage, and by errors originating in the electrical or the communication network.

Keeping functionalities in an abstraction layer, as provided by virtualisation, wherever possible, has many advantages, especially for critical infrastructure such as smart grids. The first and most obvious benefit is additional flexibility, as resources can be associated with the logical nodes on demand. Keeping software functions in defined and bounded boxes (container-based virtualisation) additionally provides better security handling, as the access to the containers can be restricted in every desired way.

Furthermore, orchestration tools for the cooperation of these containers are available (e.g., kubernetes). That way, it is possible to manage large-scale software rollouts, replication of services, upgrades of services and even the topology-aware distribution of components. Thus,

use-cases like commissioning with high requirements regarding scalability profit immensely from these virtualisation technologies.

*5.3. Overlay Networks*

Whereas historically, many critical infrastructures relied on dedicated ICT resources, the ongoing convergence of information technology with operational technology (IT/OT convergence) led to mixed systems, wherein dedicated parts interact with public ICT infrastructure. This includes public cloud systems, but also public IP connections to these cloud services or to customer premises domains, etc. Obviously, security, performance and dependability are issues for such hybrid systems.

This is especially true for low-voltage grids, where IP-based network access often has limited bandwidth. However, most network operators and energy providers are highly distributed, requiring an adequate communication infrastructure between different sites. Finally, third party organisations, which provide support for applications such as billing or self consumption optimisation, will need an efficient and secure channel to customers and to operators.

This results in conflicting requirements: Resources must be publicly available, but the access takes place in a secure and distinct way. The usual solution for this conflict is the introduction of overlay architectures [30]. A private overlay network (with restricted and controlled access) uses a public underlay infrastructure, which provides the basic connectivity. This wide-spread architectural pattern provides an abstraction of the physical infrastructure and is thus per se a virtualisation technology.

Practical applications include virtual private networks (VPNs), which can, for instance, provide secure remote connectivity, or VxLANs, which can provide a site-to-site coupling of several VLANs with distinct IP ranges. The key benefit here is the strict separation of traffic flows. If quality of service (QoS) aspects such as latency, packet loss and throughput are also to be taken into account, this can be provided by MPLS.

With MPLS, distributed applications in critical infrastructure, such as low voltage grid control, can be deployed, as MPLS provides sufficiently short reaction times for control tasks. However, for most use-cases considered in this research work, a third aspect besides separation of traffic flows and QoS provision is important: the flexibility regarding system configuration and re-configuration. This is provided by SDN.

In the context of overlay networks across existing (legacy) network infrastructure, a piece of technology called software defined wide area network (SD-WAN) [31] has been developed. Usually, SDN-capable routers provide the interface between the legacy network and the SDN world. The purpose of these devices is to provide an abstraction of the legacy network towards the SDN control plane. As a result, changes to the legacy network, such as different paths, routes or technology, remain transparent to the SDN layer. Technically, the abstraction and transparency are achieved by packet encapsulation when traffic is sent across the underlay.

Furthermore, changes made to the overlay network do not have an effect on the underlay network. For example, the implementation of new network protocols is possible on the overlay network and these protocols do not need to be implemented by the underlay network. This also allows for simplified introduction of new applications within the company network.

A drawback of SD-WAN and of all other overlay solutions is that it relies on the underlay network and only provides (very) limited monitoring or configuration capabilities of the underlying network (network as a service). This can potentially result in violations of the service level agreement. In this context, technologies such as network slicing [28] may gain high interest, as they promise to provide an underlay network with configurable service level agreements.

## 6. Case Study Implementation and Analysis

Our general model described in Section 3 suggests a large number of applications that can be based on the virtualisation control plane (see Figure 1). In order to support the model, we studied the design, implementation and evaluation results of three use-cases for virtualisation in power system

control networks. The selection of the use-case was done from the perspective of electricity distribution system operation and from the requirements that arise through the integration of renewable energies into the existing power grid. The principle of the solution in all three cases was to make use of additional functionality provided by the virtualisation framework applied (e.g., SDN Controller) and with this simplify the logic of the application.

### 6.1. Application Protocol Independent Redundancy

This section contains the main motivation, the solution and the most important results using the concepts described by von Tüllenburg et al. [13].

Looking at application-protocol-independent redundancy, we focus on failover processes for remote control and monitoring systems in electrical distribution systems. Critical components in power system control systems are usually redundantly deployed. In case of component failures, defined failover processes ensure a reliable switch-over to a working unit within a given time constraint. The novelty of the proposed approach is the use of SDN for the detection of system failures and for performing the actual switch-over.

Typically, remote control and monitoring in power distribution systems is deployed across two levels (according to the smart grid architecture model—SGAM): the station level and the operation level. The operation level groups the infrastructure, which is necessary to enable the control of field facilities using central control systems. The operation level particularly includes communication networks and the SCADA system. The station level, in comparison, denotes the equipment within field stations, which is necessary for remote control and monitoring. Besides, a communication network, at station level; IEDs; and remote terminal units (RTUs) are deployed, implementing particular functions of the remote control and monitoring system.

The gateway device is responsible for the interconnection of station level and operation level and thus is highly important for remote control and monitoring. This device is in the focus of the redundancy management in this use-case.

Figure 5 depicts three different situations for the interconnection of SCADA and field stations. In the left scenario, only one gateway device is used for the connection of field controllers (RTUs or IEDs) to the central SCADA system. This leads to a situation in which a failure of the gateway renders the field station unavailable to monitoring and control applications (a typical single point of failure). The state-of-the-art approach (shown in the centre) uses redundancy and failover management of gateway devices using a special voter component. The voter selects one of the available gateways based on its configuration with regard to predefined performance indicators. Whenever the alternative gateway performs better under these parameters, the voter switches to the other device. An example for such a parameter would be the communication quality between gateway and SCADA.

The redundant gateway approach, however, suffers from an intersection of concerns, as the voter (which is basically a part of the SCADA system) needs to be aware of the internal states of gateways, which are mainly communication devices such as switches and routers. Thereby, artificial dependencies between the SCADA system, the communication network, the communication protocols and the failover operation are created.

By contrast, the application-protocol-independent redundancy approach depicted on the right side enables a separation of concerns by use of SDN. Here, SDN is employed to monitor the communication behaviour between the Gateway devices and the SCADA systems. Thereby, gateway failures (non-operational or non-responding gateways) can be detected and an immediate switch to another available gateway can be performed without notifying the SCADA application. However, after switching to another gateway, the application layer communication (e.g., IEC 60870-5-104) needs to be re-established, as the state of the particular application layer connection between gateway and SCADA cannot be monitored using SDN. An evaluation of this approach showed that the SDN approach is able to complete the failover procedure within five seconds of failure occurrence (software and hardware failures of the gateway). This is considered sufficiently fast for most applications in field

station monitoring and control. Furthermore, arbitrary application level protocols are supported, which ensures flexibility for future scenarios.

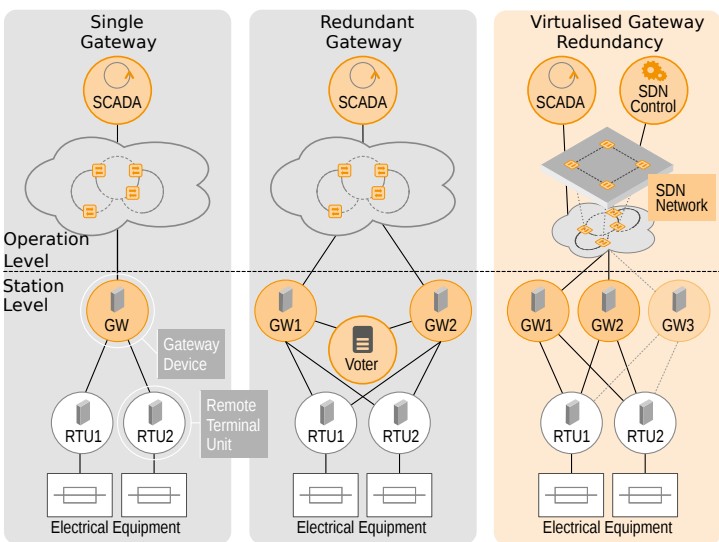

**Figure 5.** The virtualised gateway redundancy approach improves the flexibility of gateway redundancy implementations while additionally ensuring the separation of concerns.

For the implementation of the gateway-failure and switch-over operation, we had a look at two different technologies, the SDN reference implementation OpenFlow (Switch Specification, Version 1.5.1) and the network programming language P4. First, we conducted a theoretical comparison of both technologies. While both technologies allow a fast detection of broken physical network connections of the gateway (broken link), things become more complex when detecting non-responsive gateways. With OpenFlow, switch port metrics (byte and packet counts) are available to analyse the communication activity between a gateway and the SCADA system. Hence, to detect an unresponsive gateway, the 104 requests and replies sent between the main component and the gateway are analysed by observing the relation of the associated traffic, and as soon as SCADA requests are not answered by the gateway, a switch-over is performed. With P4, which in principle may be used to access the application payload of network packets, it would be possible to extract the semantic information from the 104 header. That way, request and reply messages could be semantically matched, which could improve unresponsive gateway detection reliability. However, as in common real-world implementations, the 104 traffic is usually transport layer security (TLS) encrypted, P4 cannot play to its strengths, and we decided to use an OpenFlow approach for the implementation in this project.

### 6.2. Grid Based Routing

In an ICT-enabled power system we find two types of networks, the communication network and the energy grid. Even though the communication network is often built alongside the electrical network via power line communication or via fibre cables attached to the power line, the structure of these two networks cannot be assumed to be identical or even match each other. Figure 6 shows the clusters in which data are exchanged between components. The components within a cluster are computed dynamically by grouping components with relevant information for each other at a given time depending on the current state of the energy network. e.g., whether the measurements of the voltage sensor $U_3$ have relevance to the transformer $T_1$ depends on the state of breaker $S_3$. If $S_3$ is closed, the sensor data from $U_3$ are also relevant for transformer $T_1$ in order to adapt voltage induction. Since there is no physical dependency between measurements on $U_3$ and actions taken on $T_1$, if switch $S_3$ is open, the sensor data is only relevant to transformer $T_2$.

In case of a change to the power network (changes in the topology, additional prosumers, changing power demands or partial outages), a reconfiguration of the communication flow might become

necessary. On one hand, this reactive (re-)configuration effort is quite high and prone to errors, but on the other hand, when the distribution and filtering of the available data are done at the application layer, a lot of information about the current state and the connectivity of the grid has to be provided to the components, which increases the engineering effort and the application's complexity.

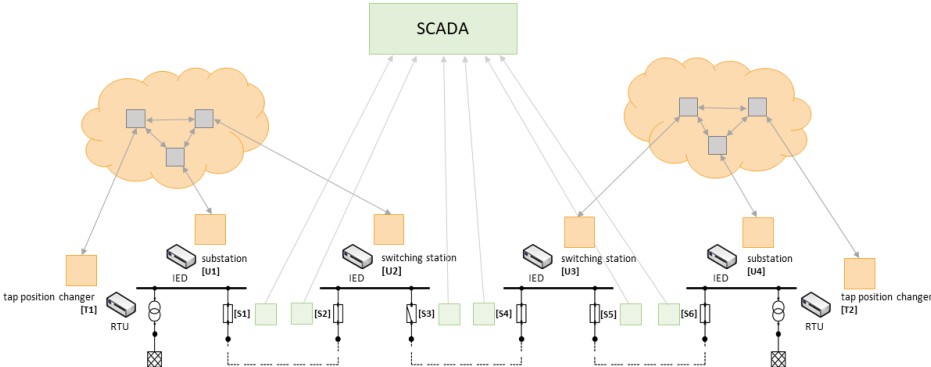

**Figure 6.** Example medium voltage grid. $T_i$ are tap position changing transformers, $S_i$ are circuit breakers and $U_i$ are metering points.

When using virtualisation technologies, this complexity can be shifted from the applications to a virtualisation layer. The application provides its data to and receives its data from a single (virtual) endpoint. Thus, the way information is distributed between the components is completely irrelevant to the application, reducing complex communication problems at the application layer to less complex point-to-point connections. A change in the energy grid as well as in the communication topology can be applied at the virtualisation subsystem level and remains completely transparent to the application, thereby also decreasing the engineering effort during commissioning.

We implemented a proof-of-concept based on SDN and a cloud or broker-based approach. Therefore, as shown in Figure 7, we split the problem into three subsystems. The application subsystem (i.e., calculating a tap position or monitoring the voltages on the line), the communication subsystem which builds the needed abstraction for rerouting the information completely transparent to the application (based on the applied communication topology) and the decision subsystem, which keeps track of the current state of the grid and maps the grid topology to the communication topology.

In the first case, using SDN, we applied additional SDN switches as ingress gateways to the communication network. Thus, the application forwards all its data to the IP address used by this ingress device. Upon receipt, the ingress gateway duplicates the packet if needed, rewrites the destination address of the packets according to the flow configuration and forwards the packets. While this approach is completely transparent to the application level, and, with the exception of the ingress switch, no new hardware has to be deployed in the field, it has a major drawback when duplication of the packets is needed. When single sender/multiple receiver scenarios are desired, connection oriented protocols (such as TCP, including TLS/Secure Sockets Layer (SSL)) are not feasible without special treatment at the application subsystem.

The second implementation utilises an Message Queuing Telemetry Transport (MQTT) based framework for packet routing. The application sends its IEC 60870-5-104 messages to an ingress gateway which translates the protocol specific messages to JavaScript Object Notation (JSON) payloads and publishes them to an endpoint specific topic. As a counterpart, an egress gateway subscribes to an application-specific topic and re-translates the JSON payload back to IEC 60870-5-104 messages, and then forwards them to the receiving application. The routing component within this system subscribes to all ingress topics and dynamically re-publishes them to the matching egress topics according to the actual grid state.

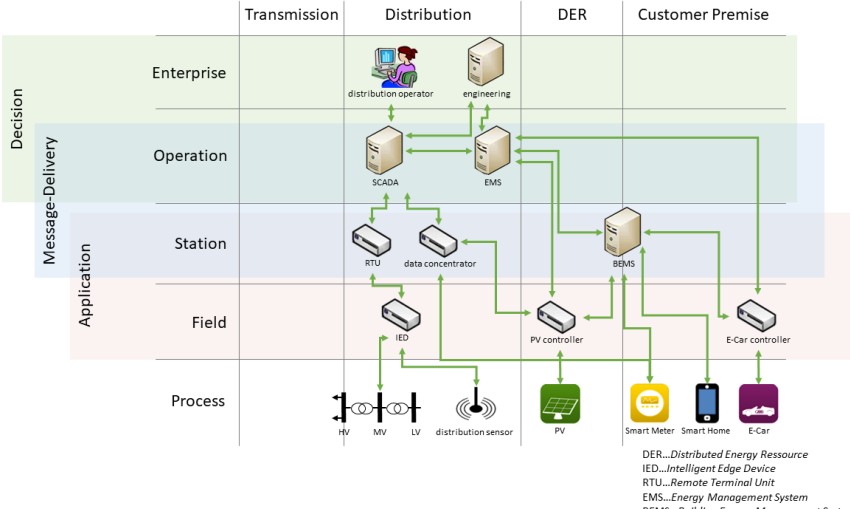

**Figure 7.** Mapping of subsystems to the smart grid architecture model (SGAM).

In comparison to the SDN approach, this approach supports single sender/multiple receiver scenarios for connection oriented protocols, and the rollout and scalability with respect to the number of endpoints is easier (no additional hardware has to be installed and all components can be managed centrally). Its major drawback is the poorer performance with respect to latency and throughput. This is caused by the fact that all routing decisions have to be made at the application layer 7 of the Open Systems Interconnection (OSI) model instead of layers 2 or 3. While the second approach supports single sender/multiple receiver scenarios for connection oriented protocols (and thus message encryption based on TLS/SSL or tunneling), true end to end security is not supported, since all the traffic has to be interpreted by the MQTT broker's routing component.

*6.3. Online System Integrity Control*

One of the main benefits of SDN is the visibility of all switches connected to the SDN controller. SDN controllers such as the popular Open Network Operating System (ONOS) controller poll switch for the flow tables in a regular manner. In the case of ONOS, flow information is requested every five seconds by default. This information is provided at high accuracy and includes information about all flows existing in the switches, providing an ideal data source for security related analyses.

Flow information is widely used for detecting security incidents. Examples for these incidents include denial-of-service (DoS) attacks and IP address or port scanning activities. The main drawbacks of flow-based analysis compared to packet-based analysis are the increased numbers of false positives and negatives due to the lower resolution of flow data.

Anomaly-based intrusion detection systems (IDS) can detect attacks by identifying deviations from normal traffic patterns. In contrast to signature-based detection systems, they are also able to detect unknown attacks. The main shortcoming of anomaly-based detection is the relatively high number of false positives. These are incidents that are classified as attacks, but that are in fact regular traffic.

Anomalies caused, e.g., by cyber attacks or faults in the network have in common that they change the distribution of IP packet header fields such as source and destination addresses or TCP ports. In the anomaly detection context, these fields characterising the traffic behaviour are the traffic features. In case of the DoS, e.g., a flooding attack, it can be expected that the number of packets directed to the victim will rise and thus change the traffic feature distribution. Additionally, a port scan will manifest in a feature distribution where a single host scans various destination IP addresses and ports from a single IP address and port.

There are numerous approaches for flow-based anomaly detection. We have applied deep neural network based algorithms, also known as Deep Learning [32]. The system consists of a classifier, an autoencoder, and a pre- and post-processing stage [33].

During pre-processing, the flow entries of a pre-defined time period are used to determine the network state. Flows are subdivided into batches, comprising of a fixed number of flow entries that are subsequently used to build the so-called partial network state (PNS). Each PNS consists of 45 features that have been selected in such a way that they represent typical network behaviour. Examples for features are IP address entropy and TCP port entropy.

The PNS is fed to the classifier and the autoencoder, as can be seen in Figure 8. The classifier performs a multi class anomaly detection by classifying the kind of attack as well as regular traffic flows. The role of the autoencoder is primarily to reduce the number of false positives. An autoencoder provides only one class anomaly detection capabilities by differentiating between regular and attack traffic.

In the post-processing stage, the results from the classifier and the autoencoder are combined in order to build the final classification. Depending on the reconstruction error that is estimated by the autoencoder, the classifier output is manipulated in order to reduce the number of false positives.

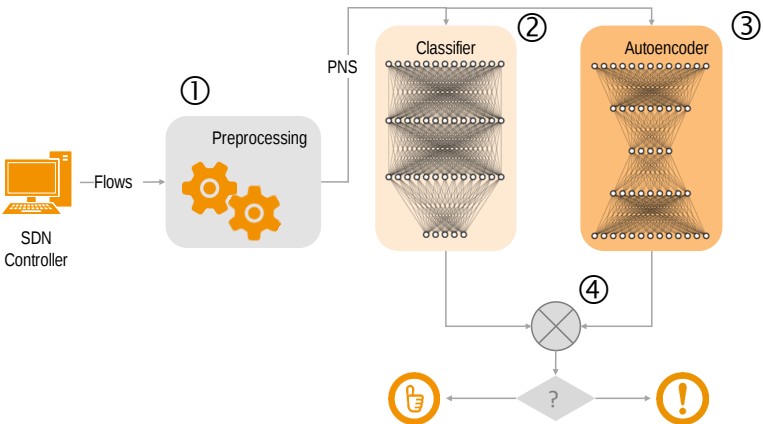

**Figure 8.** Schematic architecture: (**1**) pre-processing module, (**2**) classifier, (**3**) autoencoder, (**4**) aggregation of both models and final prediction.

As there was no flow information of a real-world SDN available, we used two non-SDN traffic captures to evaluate our system: (a) the synthetic CICIDS 2017 data set [34] for evaluating the performance of intrusion detection systems and (b) a traffic snapshot from a SCADA network using IEC 60870-5-104 traffic. As the SCADA traffic dataset did not contain any attacks, they were added later using attack traffic from the CICIDS 2017 data set. Both datasets were reduced in a first step to flow information including the seven features (source IP and TCP port, destination IP and TCP port, number of packets, number of bytes and lifetime of a flow) that are available in SDN.

For both data sets, our system showed quite good performance. For the CICIDS 2017 data set, we gained a false positive rate of only 0.57% and a true positive rate of 95.19%. Using the SCADA data set, the detection rate reached 100% with no false positives.

We could show that with the metrics derived from OpenFlow, flow statistics can efficiently be used to identify changes in the distribution of traffic features in order to identify anomalies.

However, for including specific traffic features in the flow statistics, it is necessary to perform flow matching on these features. Feature extraction is limited to layer-2 to layer-4 information. In order to gain access to application layer features, deep packet inspection using P4 was evaluated.

As mentioned earlier, P4 programming can bring quite some benefits to software defined networks. However, P4's capabilities have so far been rarely used to access the entire payload of SCADA packets.

Thus, we tried to understand to what extent it is feasible to use P4 to access and modify any part of the payload of IEC 60870-5-104 packets for the purposes of an intrusion detection system.

Having this goal in mind, a P4 parser was implemented to extract the fields from IEC 60870-5-104 packet payloads and several intrusion detection mechanisms were implemented to demonstrate the functionality of the parser and the advantages and disadvantages of using P4 for security purposes when compared to alternatives such as Snort. The evaluation of the implemented solutions showed that, despite working as intended, both parsing and intrusion detection in P4 introduced large reductions in throughput when compared to solutions that did not parse IEC 60870-5-104 payloads. However, it needs to be emphasised that the implementation relied on the Behavioural Model 2 which is supposed to provide significantly reduced performance over P4 hardware platforms such as specialised P4 capable switching hardware.

## 7. Results and Analysis

This work started with the hypothesis that virtualisation techniques can significantly reduce the complexity of use-case implementations on the application plane, as shown in Figure 1, and thus also reduce the engineering effort required.

With the implementation of the three described use-cases, deep insights into the application of virtualisation techniques in the context of power system control systems have been gained. The initial expectations of reduced complexity and engineering effort have been met. The technology readiness of the applied technologies spans from commercially available SDN solutions that can be applied straight away, to software implementations of P4 that show a principle of a solution but need more work in the underlying libraries and regarding performance.

### 7.1. Contributions

We have argued that the engineering effort for implementing use-cases on the application plane can be reduced using virtualisation techniques because the control-plane layer encapsulates the complexity of the lower-level systems (data plane and physical power grid). With the three use-case implementations, we now can support this claim.

*Application protocol independent redundancy*: The novelty of the proposed approach for this use-case is the application of SDN for the detection of system failures and for performing the actual switch-over from a failed to a redundant unit in a power system control network. Usually, the switch-over from one controller to the other is implemented using lengthy hand-coded rule sets based on dedicated parts of the IEC 60870-5-104 protocol. By implementing an application-protocol-independent solution using OpenFlow, we can significantly reduce the required effort. The engineer in charge of the deployment of the redundant system merely needs to configure the linear, non-redundant communication path from the sensor, over the controller, to the actuator. Then, on SDN-level, some of the components of this part need to be marked as redundant. The remaining logic is implemented with SDN, not requiring more application-level configuration. Complexity in terms of redundant setups is moved from the application plane to the control plane and therefore encapsulated.

*Cross-layer optimisation and grid-based routing*: We have compared the benefits of different virtualisation approaches for this application (brokerage, OpenFlow, P4). Physical power grid and data plane represent two different layers in the virtualisation model (Figure 1), and our proposed solutions improve performance across both. We showed alternatives for how to route measurement data to the appropriate destinations based on the actual grid state without any reconfiguration of the measurement devices. In comparison to the SDN solution, the cloud-based approach supports single sender/multiple receiver scenarios for connection oriented protocols. With this, it better integrates with existing SCADA systems where this scenario is often relevant. Complexity in terms of power system feeder configuration is moved from the application plane to the control plane and therefore encapsulated.

*Online system integrity control*: We have studied the implementation and associated restrictions for domain-specific anomaly detection in power system control networks when using the northbound interfaces made available by SDN controllers. One of the main benefits of SDN is the visibility of all switches connected to the SDN controller. SDN controllers such as the popular ONOS controller poll switches for the flow tables in a regular manner. Flow information is widely used for detecting security incidents. The main drawback of flow-based analysis compared to packet-based analysis is the increased number of false positives and negatives due to the lower resolution of flow data. We could show that with the metrics derived from OpenFlow, flow statistics can efficiently be used to identify changes in the distribution of traffic features in order to identify anomalies. However, for including specific traffic features in the flow statistics, it is necessary to perform flow matching on these features. Feature extraction is limited to layer-2 to layer-4 information. In order to get access to application layer features, deep packet inspection using P4 was evaluated. Complexity in terms of communication flow sensing is moved from the application plane to the control plane and therefore encapsulated.

## 7.2. Research Questions Answered

This work analysed the research questions defined in the methodology section. Expected advantages of virtualisation approaches were, first, a reduction of configuration effort by encapsulation of complexity into virtualised network layers; second, better reliability achieved by improved abilities to re-locate functionality; and third, better system state awareness. Table 1 gives an overview of the results and answers gained from the study case implementations.

**Table 1.** Answers to the research questions gained from the use-cases studied.

| Research Question/Use-case | Research question 1: Is configuration effort and manual engineering in patch-management reduced? | Research question 2: Can system reliability be improved and graceful degradation be realised? | Research question 3: Is situational awareness in power system ICT networks improved? |
|---|---|---|---|
| **Use-case 1** Application protocol independent redundancy | Configuration effort and complexity is reduced by abstracting address information. With the SDN-based solution developed in this work it is no longer necessary to configure redundant gateways, since the voter function can be integrated at network level. | Using SDN, functionality is virtually re-located to a neighbouring component. In fact, underlying communication paths are re-configured when required. The resulting solution is fail operational, switch-over times are bounded to 5 seconds, which is sufficient for SCADA applications but not for real time control. | An SDN-based solution for this problem makes use of SDN standard interfaces to detect failures in connectivity. Further analysis of communication behaviour can be performed using P4 in a non-encrypted case. |
| **Use-case 2** Cross-layer optimisation and grid-based routing | Not in focus. | Communication relations can be changed in an application-agnostic way both with SDN or brokerage solutions. In this use-case, the broker solution represented a better architectural fit to SCADA environments. Migration of functionality can be realised using application orchestration with tools such as Kubernetes or K2. | Not in focus. |

Table 1. *Cont.*

| Research Question/Use-case | Research question 1: Is configuration effort and manual engineering in patch-management reduced? | Research question 2: Can system reliability be improved and graceful degradation be realised? | Research question 3: Is situational awareness in power system ICT networks improved? |
|---|---|---|---|
| **Use-case 3** Online system integrity control | Security policies can be centrally managed when using SDN controllers, which simplifies re-configuration and commissioning of new nodes during runtime of the system. New nodes can be automatically detected and policies for network segmentation as well as access control can therefore be better enforced. | Not in focus. | By polling the flow statistics from network switches, an SDN controller has timely visibility of network states, overload conditions, errors and attacks. It was possible to fully base the anomaly detection implemented in this work on this data. |

## 8. Conclusions

This work has shown the role that virtualisation approaches can play in reducing system complexity and configuration effort when setting up and operating communication networks for energy infrastructure. Automation technology is playing an increasingly important role in the operation of electrical distribution networks. However, increasing use of ICT systems and communication links also means increases in system complexity and engineering effort during construction, maintenance and operation. Virtualisation technologies can help to manage this increasing system complexity more easily. By implementing three concrete use-cases, it was possible to draw general conclusions about the benefits (and limits) of virtualisation approaches in the given context. In particular, system operators who also operate other infrastructures such as public transport, district heating or gas networks and electricity networks, are faced with the challenge of bringing large quantities of new communication endpoints "into the area" for connecting additional sensors and actuators.

Already at the design time of automation applications, the use of SDN, for example, allows functionalities such as redundancy provision and redundancy activation to be encapsulated. They therefore no longer need to be considered at the application level in individual cases, but are automatically available as a communication level capability.

When the system is in operation, SDN or cloud approaches allow the system complexity to be reduced, for example, by adapting the communication paths either on the network level or in a central (possibly redundant) broker, so that sensor values are always dynamically linked to the correct controller. In addition, the system integrity of the ICT network, but also indirectly the integrity of the electricity network, can be monitored more easily if the standardised flow monitoring interfaces of SDN controllers are used.

The implementation of virtualisation concepts in existing networks and automation systems can be realised with comparatively little effort. Not all routers need to be upgraded to enable SDN; only the devices on either end of the communication path need to be replaced by SDN-capable variants. Broker solutions are well suited to easily expand existing SCADA systems.

**Author Contributions:** F.K. contributed to the introduction, defined the research questions and method, contributed to the results and analysis section and gave input to use-case definition. He defined the paper structure and is the corresponding author. A.V. contributed to the overall structure of paper; edited Section 5 and the main part of Section 6.1; and proof read the whole paper. F.v.T. contributed to the related work (Section overview and Section 4.1), contributed to the use-case grid-based routing (Section 6.2) and mainly edited the section covering the use-case application-protocol-independent redundancy (Section 6.1). P.D. contributed to the use-case application-protocol-independent redundancy and to the general structure of the paper. U.P. contributed to the state-of-the-art Section 4.2 and the definition and implementation of the test environments of the use-cases application-protocol-independent redundancy (Section 6.1) and grid-based routing (Section 6.2). He also edited Section 5.3. A.H. contributed to the use-case section about grid-based routing (Section 6.2), the design and implementation of the prototypes used within this use-case and the general structure of the paper. O.L.

contributed to the use-case application-protocol-independent redundancy (Section 6.1) and performed proof reading of the entire document. O.J. contributed to online system integrity control (Section 6.3) and the system model (Section 3). R.F. contributed the state of the art on P4 (Section 4.3). All authors have read and agreed to the published version of the manuscript.

**Funding:** This project was funded by the Austrian Climate and Energy Fund and was implemented as part of the Energy Research Program 2016.

**Conflicts of Interest:** The authors declare no conflict of interest.

## Abbreviations

The following abbreviations are used in this manuscript:

| | |
|---|---|
| AMI | Advanced Metering Infrastructure |
| BEMS | Building Energy Management System |
| DER | Distributed Energy Resources |
| DoS | Denyal of Service |
| EMS | Energy Management System |
| GOOSE | Generic Object Oriented Substation Event |
| ICT | Information and Communication Technology |
| IDS | Intrusion Detection System |
| IED | Intelligent Electronic Device |
| IP | Internet Protocol |
| IT | Information Technology |
| JSON | JavaScript Object Notation |
| LAN | Local Area Network |
| LV | Low Voltage |
| MBSE | Model-Based System Engineering |
| MDA | Model-Driven Architecture |
| MDA | Model-Driven Engineering |
| MPLS | Multi-Protocol Labal Switching |
| MQTT | Message Queuing Telemetry Transport |
| MV | Medium Voltage |
| ONOS | Open Network Operating System |
| OSI | Open Systems Interconnection |
| OT | Operational Technology |
| P4 | Programming Protocol-independent Packet Processors |
| PDC | Phasor Data Concentrator |
| PMU | Phasor Measurement Unit |
| PNS | Partial Network State |
| QoS | Quality of Service |
| RTU | Remote Terminal Unit |
| SCADA | Supervisory Control and Data Acquisition |
| SDECN | Software-Defined Energy Communication Network |
| SDN | Software Defined Network |
| SD-WAN | Software Defined Wide Area Network |
| SGAM | Smart Grid Architecture Model |
| SSL | Secure Sockets Layer |
| TCP | Transmission Control Protocol |
| TLS | Transport Layer Security |
| TSN | Time Sensitive Network |
| VLAN | Virtual Local Area Network |
| VM | Virtual Machine |
| VPN | Virtual Private Network |
| VxLAN | Virtual (Extensible) Local Area Network |
| WAMS | Wide Area Monitoring Systems |

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
