# Peer review of "The Impact of Virtualisation Techniques on Power System Control Networks"

_electronics, doi:10.3390/electronics9091433_

Round 1

Reviewer 1 Report

It is a very interesting topic concerning the impact of virtualisation techniques in power system control networks.  This work could be published as long as the authors make the following changes:

a) The state of the Art and what has already been done is very extensive. On the other hand the innovation of this work is not clear and should be further analyzed.

b) Authors should present more clear the contribution of their work in the "Results and Analysis" section

Author Response

Dear Reviewer,

thank you for your comments. We have strongly reworked the paper. As suggested, we clearly state the contributions made now in the "Results and Analysis" section.

We have further improved the clarity of the innovation of this work by introducing a system model (new section 3) and stating our research hyothesis based on that model. We then support the hyphothesis by three implemented use cases.

With this, we intended to address also the research design and the clear desicription of methods, results and conclusion.

We have removed the Commissioning use case, since it indeed was lacking a clear solution methodology as well as quantitative results. This was due to the delay of a field trial, of which we had expected further material for the paper but finally had to live without that.

Kind Regards

Friederich Kupzog

Reviewer 2 Report

The article “Impact of Virtualisation Techniques in Power System Control Networks” addresses the virtualisation techniques in power system. However, this paper needs to improve significantly in method development and writing.

The Author should clarify and justify the main contributions of this paper.

The paper should have a robust model for the virtualization of the power system control network, and after that, you may adopt the case study with the proposed model.

The abstract has an unnecessary background, and it should be concise with problems and proposals.

The readability of the paper is very low. The authors should improve the writing of the whole paper with correct sentence structure and massive grammatical correction.

Author Response

Dear Reviewer,

thank you for your comments. We have strongly reworked the paper. Please find below the measures according to your suggestions.

(1) The Author should clarify and justify the main contributions of this paper.

-> As suggested by another review, we clearly state the contributions made now in the "Results and Analysis" section.

(2) The paper should have a robust model for the virtualization of the power system control network, and after that, you may adopt the case study with the proposed model.

-> This in particular was a helpful suggestion and also has been made by a second reviewer. We have introduced a system model (new section 3) and now state our research hyothesis based on that model. As suggested, we then support the hyphothesis by three implemented use cases.

(3) The abstract has an unnecessary background, and it should be concise with problems and proposals.

-> We have removed this unnecessary background and improved the readability of the abstract.

(4) The readability of the paper is very low. The authors should improve the writing of the whole paper with correct sentence structure and massive grammatical correction.

-> we have reworked the language of the paper in several rounds and also had a deep review made by a native English speaker.

In order to further improve the quatlity of the overall research design and the clear desicription of methods, results and conclusion, we have removed the Commissioning use case, since it indeed was lacking a clear solution methodology as well as quantitative results. This was due to the delay of a field trial, of which we had expected further material for the paper but finally had to live without that.

Kind Regards

Friederich Kupzog

Reviewer 3 Report

Authors analyze in this paper how contemporary virtualization approaches can support in electricity grid management and control systems set up and development.

From my point of view, it is not clearly demonstrated enough how virtualization techniques can support operators with the integration of large numbers of new communication nodes. The orchestration workflow between the different resources involved should be more detailed. Then for instance, the commissioning concrete use case (5.1) is not sufficiently commented and no clear conclusion is given on how performance increase and how they are measured.

The authors give an introduction about commissioning (here a reference is required), application protocol independent redundancy, grid-based routing and online system integrity control. They conclude that virtualization contributes to reduce engineering effort, better system management, and simplification in high-level system architecture. Nevertheless, from a methodological point of view it is not clear how the virtualization approach hide the implementation details and how it can be applied and replicated. I suggest to read and refer to documents from System Engineering domain (in particular INCOSE wrote convenient documents about this subject). For instance, MBSE can provide interesting Model based approach to pave the way from concepts to implementation of systems.

The result and analysis section (6) should better conclude on why enabling software-defined networking for instance, especially by giving more quantitative argument to explain why only key routers have to be exchanged.

As a conclusion, the paper is not yet ready to be published since the virtualization technique used, integrating network virtualization into an existing control network has to be more methodologically presented (pedagogically as well) to understand how low effort is required using it.

Author Response

Dear Reviewer,

thank you for your comments. We have strongly reworked the paper. Please find below the measures according to your suggestions.

(1) From my point of view, it is not clearly demonstrated enough how virtualization techniques can support operators with the integration of large numbers of new communication nodes. The orchestration workflow between the different resources involved should be more detailed.

-> We have improved the presentation of the research design in particular by follwing your suggestion in Point (3) and (4).

(2) Then for instance, the commissioning concrete use case (5.1) is not sufficiently commented and no clear conclusion is given on how performance increase and how they are measured.

-> We have removed the Commissioning use case, since it indeed was lacking a clear solution methodology as well as quantitative results. This was due to the delay of a field trial, of which we had expected further material for the paper but finally had to live without that.

(3) The authors give an introduction about commissioning (here a reference is required), application protocol independent redundancy, grid-based routing and online system integrity control. They conclude that virtualization contributes to reduce engineering effort, better system management, and simplification in high-level system architecture. Nevertheless, from a methodological point of view it is not clear how the virtualization approach hide the implementation details and how it can be applied and replicated. I suggest to read and refer to documents from System Engineering domain (in particular INCOSE wrote convenient documents about this subject). For instance, MBSE can provide interesting Model based approach to pave the way from concepts to implementation of systems.

-> This in particular was a helpful suggestion and also has been made by a second reviewer. We have introduced a system model (new section 3) and now state our research hyothesis based on that model. We then support the hyphothesis by three implemented use cases.

(4) The result and analysis section (6) should better conclude on why enabling software-defined networking for instance, especially by giving more quantitative argument to explain why only key routers have to be exchanged.

-> As also suggested by another review, we have strongly reworked the Results and Analyses section (now Chapter 7) and clearly state the contributions made.

(5) As a conclusion, the paper is not yet ready to be published since the virtualization technique used, integrating network virtualization into an existing control network has to be more methodologically presented (pedagogically as well) to understand how low effort is required using it.

-> We have addressed the methodolodic presentation of our results by intoducing the new system model in Section 3, and now state our research hyothesis based on that model. We then support the hyphothesis by three implemented use cases.

-> We have reworked the pedagicical presentation of the content in several rounds and also had a deep review made by a native English speaker.

Kind Regards

Friederich Kupzog

Round 2

Reviewer 2 Report

Authors addressed all the concerned and it can be accepted for publication.

Author Response

Thanks again for the feedback. Kind Regards Friederich Kupzog

Reviewer 3 Report

I found that most of my comments have been considered.

Nevertheless the section dedicated to Model Based and/or Model Driven going as I suggested previously bu the following sentences:

"I suggest to read and refer to documents from System Engineering domain (in particular INCOSE wrote convenient documents about this subject). For instance, MBSE can provide interesting Model based approach to pave the way from concepts to implementation of systems."

These comments have not been fully considered. I invite the authors to improve again this section. More reference papers MBSE, MDA, MDE, need to be cited here as well. How to pass from Models to implementation with reducing the risk of devellopping wrong system behavior.

As well model ans system validation should be discussed. How can you stipulate that your model and system is sound enough when you provide models ?

Then I recommend asking for English proofreading of the paper since some sentences are still not correctly structured.

Author Response

Dear Reviewer,

thanks again for pointing to your initial suggestion "I suggest to read and refer to documents from System Engineering domain (in particular INCOSE wrote convenient documents about this subject). For instance, MBSE can provide interesting Model based approach to pave the way from concepts to implementation of systems."

We agree with your suggestion that using MBSE and especially MDE is a potential and attractive research design option for the problem presented in our work. In response to your suggestion, we have added references 7...12 and a discussion on the choice of the methodology to chapter 2. I repeat it here, since it can be read also as a direct answer to your remark.

"A research design appropriate to answer the research questions can be chosen in different ways. One option is the use of a model-based system engineering (MBSE) approach [7]. In MBSE, a formal model of the system of interest is developed and can be used for code generation, test case generation, or automatic verification and more. This approach would allow a structural evaluation of the advantages of the virtualisation-based problem solution by programmatic analysis of the model. In model-driven architecture (MDA), the prime artefact of the work is actually the model itself, which is typically formulated using Unified Modelling Language [8]. Model-Driven Engineering (MDE), which makes use of domain-specific languages [9] has already been successfully applied to the power system domain already [10].

However, the research questions address a range of mostly qualitative questions. This is due the nature of the initially open question about potential advantages of virtualisation in power system control networks. Here, a formal system model based on a domain-specific description language comes with challenges. The scope and application environments of the use case implementations were unknown in the beginning of the research, and with this the structure and potential complexity of the model. Therefore, this work is based on an empirical approach. We conduct applied research and implement three study cases that allow us to generalize findings to a certain extend. With that, we establish the validity of our claims on the “accumulated weight of empirical evidence” [11]. This approach is also known as “validation by example” in the taxonomy of Shawn et al. [12]. Nevertheless, we describe our approaches and results in the context of a system model (Chapter 3)."

We hope to have made the relationship between our work and MDSE more clear, as well as the reasons why we have chosen a different approach. The discussion of the empirical approach and references 11 and 12 are in relation to your remark: "How can you stipulate that your model and system is sound enough when you provide models ?"

We further agree that the application of MBSE would be one way to reduce the risk of developing a wrong system behavior. In this work we however are comparing a "conventional" with a "virtial" solution of the same problem. By implementing and practical testing of the software artefacts, we can clearly judge correct system behaviour in comparison to the conventional solution.

In regard to incorrectly structured sentences, the paper has undergone a rigid proofreading by a native speaker in the first round. We have re-visited the grammar again and simplified some sentences in Chapters 4 and 5 as well as in the conclusion. Of course, there can always be something left that we have overseen, however we are strongly confident that the quality level of the language is now very high. If you have a specific case in mind, we would be grateful if you could point it out.

Kind Regards

Friederich Kupzog

  1. Rodrigues da Silva, A. Model-driven engineering: A survey supported by the unified conceptual model. Computer Languages, Systems & Structures 2015,43, 139–155. doi:10.1016/j.cl.2015.06.001.
  2. Estefan, J.A.; others. Survey of model-based systems engineering (MBSE) methodologies. Incose MBSE, Focus Group 2007,25, 1–12.
  3. Schmidt, D.C. Model-Driven Engineering.Computer 2006,39, 25–31. doi:10.1109/MC.2006.58.
  4. Andrén, F.P.; Strasser, T.I.; Kastner, W. Engineering smart grids: Applying model-driven development from use case design to deployment.Energies 2017,10, 374.
  5. Hevner, A.R.; March, S.T.; Park, J.; Ram, S. Design science in information systems research.MIS quarterly 2004, pp. 75–105.

12.Shaw, M. What makes good research in software engineering? International Journal on Software Tools for Technology Transfer 2002,4, 1–7

Round 3

Reviewer 3 Report

The authors answered to my last round of questions in the method chapter. Now the recall about MB and MD method is ok. In the text added, they explain that they developed an experimental study and that, if I understand well, formal models were not defined before implementation. But I don't clearly know if models (what kind?) are used or not at the end of line 100.

Then, Authors announce that they used High level models in the method description in the contribution section 5, so models even semi or not formal are used, again please clarify in paragraph starting at line 91. If so, after this chapter, I can't see clearly how Models and and MD/MB methods are exploited in section 5 and 6. These sections still need to be improved referring better to the method presented at the beginning.

Concerning my understanding, The paragraph starting line 101 with "The applied methodology therefore consists of the following steps: First, a high-level model for ..." is not well structured, where are the other steps? It is an important part of the paper methodology bit it is difficult to follow.

Author Response

Dear Reviewer,

we have considered your new suggestions as shown below.

In the text added, they explain that they developed an experimental study and that, if I understand well, formal models were not defined before implementation. But I don't clearly know if models (what kind?) are used or not at the end of line 100.

--> We have added the following explanation to the referred paragraph:

“Nevertheless, we describe our approaches and results in the context of a high-level architecture model as described in Chapter 3 to better show the similarities in the virtualisation of the three study cases.  “

Then, Authors announce that they used High level models in the method description in the contribution section 5, so models even semi or not formal are used, again please clarify in paragraph starting at line 91. If so, after this chapter, I can't see clearly how Models and and MD/MB methods are exploited in section 5 and 6. These sections still need to be improved referring better to the method presented at the beginning.

--> We are not following a MD/MB approach as such. We use the high-level architecture model as described in Chapter 3 to better show the similarities in the virtualisation of the three study cases. Therefore, I would find it misleading for the reader to discuss this matter again in chapter 5, as it is only discussing different virtualisation options.

We have therefore added the following text in the beginning of Section 6: “The principle of solution in all three cases is to make use of additional functionality provided by the virtualisation framework applied (e.g. SDN Controller) and with this simplify the logic of the application.”

Concerning my understanding, The paragraph starting line 101 with "The applied methodology therefore consists of the following steps: First, a high-level model for ..." is not well structured, where are the other steps? It is an important part of the paper methodology bit it is difficult to follow.

--> We have made the step-by-step description more clear and corrected an error here:

“First, practically relevant use cases for virtualisation in power system control networks have been identified by a team of power system operators, solution providers and researchers.

Second, a high-level model for power system control systems with implemented virtualisation techniques has been set up (this is described in Chapter 3 and shown in Figure 1). This model is a high-level architectural view and allows to generalise the virtualisation of networking functions in the given application context. We claim, that virtualisation reduces engineering effort in the practical deployment and maintenance of the identified use cases.

Third, to support the claim, for each of the three use cases, a theoretical solution has been developed and evaluated: how are planning and operational processes changed by the solution and what are the implications of these changes?

The fourth step is the experimental implementation of promising approaches in all three cases. For the evaluation, experimental test-beds in simulations, laboratory and in a real field environment have been set up, where the initial hypothesis of feasibility and impact of the virtualisation approaches could be validated.”

Kind regards

Friederich Kupzog